# Identification of Novel Antifungal Skeleton of Hydroxyethyl Naphthalimides with Synergistic Potential for Chemical and Dynamic Treatments

**DOI:** 10.3390/molecules27238453

**Published:** 2022-12-02

**Authors:** Pengli Zhang, Vijai Kumar Reddy Tangadanchu, Chenghe Zhou

**Affiliations:** 1Institute of Bioorganic & Medicinal Chemistry, Key Laboratory of Applied Chemistry of Chongqing Municipality, School of Chemistry and Chemical Engineering, Southwest University, Chongqing 400715, China; 2Zhongshan Institute for Drug Discovery, Shanghai Institute of Materia Medica, Chinese Academy of Sciences, Zhongshan 528400, China; 3Drug Discovery and Development Center, Shanghai Institute of Materia Medica, Chinese Academy of Sciences, Shanghai 201203, China; 4Department of Radiology, Washington University School of Medicine in St. Louis, St. Louis, MO 63110, USA

**Keywords:** naphthalimide, antifungal, reactive oxygen species, membrane

## Abstract

The invasion of pathogenic fungi poses nonnegligible threats to the human health and agricultural industry. This work exploited a family of hydroxyethyl naphthalimides as novel antifungal species with synergistic potential of chemical and dynamic treatment to combat the fungal resistance. These prepared naphthalimides showed better antifungal potency than fluconazole towards some tested fungi including *Aspergillus fumigatus*, *Candida tropicalis* and *Candida parapsilosis* 22019. Especially, thioether benzimidazole derivative **7f** with excellent anti-*Candida tropicalis* efficacy (MIC = 4 μg/mL) possessed low cytotoxicity, safe hemolysis level and less susceptibility to induce resistance. Biochemical interactions displayed that **7f** could form a supramolecular complex with DNA to block DNA replication, and constitute a biosupermolecule with cytochrome P450 reductase (CPR) from *Candida tropicalis* to hinder CPR biological function. Additionally, **7f** presented strong lipase affinity, which facilitated its permeation into cell membrane. Moreover, **7f** with dynamic antifungal potency promoted the production and accumulation of reactive oxygen species (ROS) in cells, which destroyed the antioxidant defence system, led to oxidative stress with lipid peroxidation, loss of glutathione, membrane dysfunction and metabolic inactivation, and eventually caused cell death. The chemical and dynamic antifungal synergistic effect initiated by hydroxyethyl naphthalimides was a reasonable treatment window for prospective development.

## 1. Introduction

Pathogenic fungal diseases account for about 60% of human and animal diseases, which have the characteristics of great harmfulness, wide spread and difficult to control thoroughly [1]. Recently, the widely used chemical agents may cause drug resistance of pathogens and form ecological hidden dangers that are difficult to predict. Therefore, it is urgent to develop novel antifungal agents with high effectivity and safety to meet the needs of survival and development of mankind. For the purpose of solving this huge challenge, it is a pragmatic tactic to discover new means to heighten the fungicidal effects [2,3]. In the methods to overcome resistance, the integration of dynamic treatment dominated by reactive oxidative species (ROS) with traditional chemical treatment may express a strategy to defeat fungi [4,5]. The effectivity of chemical drug treatment is self-explanatory, and the excess expression of ROS, the dominators of dynamic treatment, directly causes the imbalance of redox system and oxidative stress, which can trigger DNA mutation, damage cell lipids and proteins and ultimately result in cell death [6,7]. Moreover, pathological cells are more likely to be exposed to oxidative stress, so enhancing intracellular ROS levels and impairing antioxidant systems can disturb the balance of prooxidant-antioxidant environment of compromised cells and trigger cell death [8,9]. Therefore, antifungal agents that efficaciously trigger the generation and accumulation of ROS display a conspicuous battery of drug candidates worthy of further evaluation for sufferers with fungal infection in clinical trials.

Naphthalimide moiety as a unique skeleton with large tricyclic planar configuration, cycloheximide and naphthalene framework, has been supposed as a DNA-targeting chemotherapy backbone toward compromised cells [10,11,12,13]. It can intercalate into the base pair of DNA double strands, causing the double strands to rupture, which in turn affects DNA synthesis and leads to DNA damage [14,15,16]. The amido group presented in naphthalimide moiety can bind non-covalently with a variety of functional enzymes including lipase to exert antifungal activity. Modifications of naphthalimido moiety at the *N*-position and 4-position have a prominent effect on the interactions with enzymes and DNA [17,18,19]. Besides, numerous molecules containing naphthalimido moiety have been proved to be expected triggers for the production and accumulation of ROS by means of DNA damage channel, which would tremendously facilitate its application in medicinal chemical biology [20,21,22,23]. Therefore, naphthalimido moiety was considered as a promising chemical and dynamic antifungal structural backbone by manipulating supramolecular interactions and ROS regulation. Ethanol has long been applied as disinfectants in life, and introduction of hydroxyethyl fragment as hydrogen bond donor, can affect supramolecular interaction with biomolecules and might helpfully improve antifungal activities [24,25,26,27].

With respect to the foregoing, taking advantage of the structure and biochemical properties, hydroxyethyl fragment was merged into the *N*-position of naphthalimide core and the bromine atom at 4-position was replaced by amines, ethers and thioethers to afford desirable potential antifungal molecules (Figure 1). The structural properties, binding effects with DNA and antifungal activities of target naphthalimide compounds were assessed to investigate its chemicobiological behaviors. The medicinal chemical potentials of highly active compound were further elaborated, including toxicity and haemolytic assessment, ADME study, resistance development, lipase affinity, biochemical interactions with DNA and cytochrome P450 reductase, up-regulation of ROS and ROS-mediated apoptosis pathways, to explore its application possibility.

## 2. Results and Discussion

### 2.1. Chemistry

Novel naphthalimido hybrids modified by hydroxyethyl fragment were derived starting from commercial 4-bromo-1,8-naphthalic anhydride. As outlined in Figure 1 and Figure 2, the available 4-bromo-1,8-naphthalic anhydride **1** was treated with ethanolamine in the presence of ethanol to offer hydroxyethyl naphthalimido intermediate **2** with 86.7% yield. Intermediate **2** was further reacted with amines, ethers and thioethers to give the target amine derivatives **3a**–**b**, **4a**–**c** and **5**, hydroxyl derivatives **6a**–**c**, mercaptoazoles **7a**–**f** and sulfhydrypyrimidines **8a**–**d** with moderate to good yields [28,29]. The chemical structures of all novel hydroxyethyl naphthalimides were confirmed by ^1^H NMR, ^13^C NMR and HRMS spectra, and the purities were determined by HPLC spectra. In the ^13^C NMR spectra for hydroxyethyl naphthalimides, the chemical shifts around 160–165 ppm were primarily attributed to the carbons in carbonyl groups of naphthalimide backbone, while in the ^1^H NMR spectra, the chemical shifts in the range of 8.85–7.23 ppm were deemed as the aromatic hydrogens (H-Ar) fused in naphthalimide backbone. Furthermore, the HRMS results were consistent with the structures of novel hydroxyethyl naphthalimides that displayed in the schemes, and purity analysis showed that the purities of all hydroxyethyl naphthalimides were above 95%.

### 2.2. Relationship between DNA Binding and Antifungal Assay

The supramolecular interactions of the hydroxyethyl naphthalimides with DNA and their antifungal activities in vitro were further evaluated. The binding effects of compounds with DNA were measured using UV-vis spectra. All compounds exhibited outstanding binding abilities with DNA (Figure 2), which were potentially correlated with their antifungal activities (Table 1).

The activities of almost all the target compounds towards *A. fumigatus* and *C. tropicalis* were stronger than that of fluconazole. In symmetric amine series **3a**–**b**, the same antifungal values were observed, and diethylamine derivative **3b** showed higher DNA binding ability. In the hybridization of multiple hydroxyethyl fragments, derivative **4c** with three hydroxyethyl moiety exerted outstanding DNA affinity, indicating that multiple hydroxyethyl fragments were advantageous for non-covalent binding to DNA. Among mercaptoazoles modified hydroxyethyl naphthalimides **7a**–**f**, thioether benzimidazole **7f** gave better anti-*C. tropicalis* efficacy (MIC = 4 μg/mL) than fluconazole based on the antifungal activities presented, which was consistent with its excellent DNA binding ability. Similarly, sulfhydrypyrimidine **8d** in sulfhydrypyrimidine series **8a**–**d** performed remarkable DNA binding ability, and its antifungal activities shared prominent inhibitory efficacy, more potent than **8a**–**c**. Given antifungal potential of hydroxyethyl naphthalimides, thioether benzimidazole **7f** was used as model compound for farther exploration.

### 2.3. Supramolecular Interaction of Thioether Benzimidazole ***7f*** with DNA

The specific relationship between DNA and thioether benzimidazole **7f** was studied. With a fixed amount of DNA, absorption spectra were measured with increasing concentrations of **7f**. The DNA peak at 260 nm in Figure 3A proportionally disappeared with adding amount of **7f**. A weak hypochromicity between compound **7f** and DNA was demonstrated, and a slight red shift at maximum absorption wavelength was observed possibly due to the reason that the aromatic chromophore of thioether benzimidazole **7f** intercalated into the helix of DNA following the increasement of the π-π conjugation [30,31].

To expound the binding mode between thioether benzimidazole **7f** and DNA, the existing dyes both commercial acridine orange (AO) and marketable 4′,6-diamidino-2-phenylindole (DAPI) were used as spectral probes referring the reported literature [32]. As indicated in Figure 3B–D, the intensity of **7f** decreased obviously at 537 nm, which suggested that **7f** could embed into DNA by competing with AO. Moreover, the changes of fluorescence intensity of AO-DNA and DAPI-DNA with different concentrations of **7f** was compared, and it was found that the effect of **7f** on AO-DNA was stronger than that of DAPI-DNA, indicating that **7f** was mainly intercalated into DNA rather than small groove binding with DNA.

### 2.4. Cytotoxicity, Hemolysis Assays and Resistance Development Assay

The cytotoxicity and hemolysis undergoing with thioether benzimidazole **7f** were implemented to assess its underlying toxicity. Cytotoxic result showed that compound **7f** had little effect on the growth of LO2 cell line (IC_50_ = 163 μM) in the high concentration (100 μg/mL), and after exposure to compound **7f** for 1 h, hemolytic rate was lower than 5% at anti-*C. tropicalis* concentration, indicating that compound **7f** presented relative biosecurity (Figure 4A,B). These compounds could selectively target fungal cell membranes due to an electrostatic distinction on the membranes between fungi and mammalian cells [33,34]. Thus, the tendency of resistant development of **7f** against *C. tropicalis* was conducted, and fluconazole was selected as a positive control (Figure 4C) [35,36,37,38]. The MIC values of thioether benzimidazole **7f** almost remained consistent throughout the 16 passages, whereas that of reference drug fluconazole increased dramatically after the eighth passage. The result from the resistance study showed that *C. tropicalis* was unable to develop rapid resistance against compound **7f**.

### 2.5. Pharmacokinetic Properties

The online softwares PreADMET and SwissADME were performed to further research the pharmacokinetic properties and druggability of thioether benzimidazole **7f** (Table 2). The Lipinski rule, a crucial determinant in drug design and exploitation, was applied to assess theoretical pharmacological activity of thioether benzimidazole **7f** [39]. Thioether benzimidazole **7f** possessed the same bioavailability score with fluconazole and abided by Lipinski rule, which proved that **7f** equipped good pharmacokinetic properties. Besides, thioether benzimidazole **7f** displayed III category acute oral toxicity and passive response for blood–brain barrier (BBB) criteria, which indicated that compound **7f** was uninjurious for oral administration. All pharmacokinetic parameters revealed that thioether benzimidazole **7f** implemented considerable pharmacokinetic profile and outstanding drug-likeness.

### 2.6. Lipase Affinity of Thioether Benzimidazole ***7f***

Moreover, thioether benzimidazole **7f** presented strong lipase affinity, which facilitated its permeation into cell membrane. As a crucial enzyme responsible for hydrolysis of lipids, lipase widely existed in plants, animals and microorganisms. Especially, the phospholipid layer on the surface of fungi contains a large number of lipases, and antifungal agents with strong lipase affinity can more easily combine with the cell membrane. Lipase is a single spherical polypeptide composed of more than 400 amino acid residues, including seven fixed fluorescent tryptophan [40]. Therefore, when the compound binds with lipase, the physiological environment of tryptophan residues and the enzyme structure will be significantly changed, and the corresponding fluorescence intensity will be decreased (λ_ex_ = 290 nm, λ_em_ = 340 nm). As shown in Figure 5, the fluorescence intensity of lipase at 340 nm decreased with the increase in the amount of compound **7f**, indicating that compound **7f** had strong lipase affinity.

### 2.7. Membrane Damage Assay

Membrane depolarization undergoing with **7f** was explored using a fluorescent probe diSC35. The diSC35 dye entering the active cell is separated by the inner and outer membranes of the fungal cell membrane, and its fluorescence gets quenched. However, the fluorescence intensity of diSC35 dye will increase following get out of the cell if the fungal membrane is depolarized by antifungal agents. As displayed in Figure 6A, compared with the dye labeled by untreated strain, a time-dependent increase was observed in the fluorescence intensity of the dye for *C. tropicalis* treated with thioether benzimidazole **7f**, which indicated that **7f** could interact with the cell membrane of *C. tropicalis* and cause its membrane depolarization.

Moreover, the membrane permeability of *C. tropicalis* treated by thioether benzimidazole **7f** was detected through estimating the uptake efficiency of propidium iodide (PI). As a living cell membrane impenetrable dye, PI can permeate the membranes of dead *C. tropicalis* strains, but cannot enter integrated living membranes [41,42,43]. The fact of a concentration-dependent growth in the PI fluorescence verified the potential of thioether benzimidazole **7f** to cause physical destruction of the *C. tropicalis* membranes as depicted in Figure 6C. Further, the PI uptake could be visually confirmed. In Figure 6D, the red fluorescence appearance of PI dye for *C. tropicalis* incubated with compound **7f** was distinctly observed, demonstrating that compound **7f** could efficiently destroy the membrane integrity of *C. tropicalis*.

In addition to the transformation of membrane permeability, the leakage of proteins from *C. tropicalis* strains treated by thioether benzimidazole **7f** was assessed employing standard Bradford assay. The result of protein leakage from *C. tropicalis* was presented in Figure 6B. It is proof that a dose-dependent enhancement in protein leakage was observed from *C. tropicalis* treated by thioether benzimidazole **7f**, which indicated membrane damage and loss of cellular integrity for *C. tropicalis* strains.

### 2.8. Supramolecular Interaction of Compound ***7f*** with Cytochrome P450 Reductase

Cytochrome P450 reductase (CPR) (PDB ID: 6T1U) as an attractive target to investigate the antifungal mechanism was subjected into ligand–receptor docking to rationalize the observed antifungal activity and understand the possible mechanism. Compound **7f** could form a biosupramolecular complex with CPR from *C. tropicalis* by multiple hydrogen bonds and other non-covalent interactions (Figure 7). The O atom of carbonyl group at 1-position in naphthalimide was bound to H atom of amino group in SER-441 with a space distance of 1.8 Å, and the H atom of hydroxyethyl segment could interact with O atom of carboxyl group in ASP-677 with a space distance of 1.9 Å. The N atom and H atom of benzimidazole fragment took part in hydrogen bonds reciprocity with TRP-679 and GLU-460 residues with a space distance of 2.3 Å and 1.9 Å, respectively. All these non-covalent interactions indicated that compound **7f** could interact with cytochrome P450 reductase to disturb its biological function [44,45,46].

### 2.9. ROS-Mediated Dynamic Treatment

In addition to intrinsic structural advantages by supramolecular interactions with DNA and CPR, thioether benzimidazole **7f** could induce the up-regulation of cytotoxic ROS to cause inevitable impairment for cells. Additionally, thioether benzimidazole **7f**-induced ROS production on the basis of fluorometric method by 2′,7′-dichlorofluorescin diacetate (DCFH-DA) dye was evaluated [47,48,49,50]. The fluorescence intensity of DCFH-DA dye at 528 nm preincubated by *C. tropicalis* strain and thioether benzimidazole **7f**, occurred a concentration-dependent augment, which obviously inferred that thioether benzimidazole **7f** could trigger ROS accumulation in Figure 8A. Reactive nitrogen intermediates (RNIs), such as NO, ONOO^-^ and S-nitrosothiols, are similar to ROS and can eradicate pathogen tissues independently or synergistically by acting on nucleic acids, proteins or lipids of pathogen [51]. As provided in Figure 8C, the variation trend of intracellular RNIs in *C. tropicalis* strains was estimated by Griess’s reaction. It was proof from the consequences that time and dose-dependent changes in RNIs production were noticed from *C. tropicalis* treated by thioether benzimidazole **7f**. The maximum generation of RNIs in *C. tropicalis* strains was acquired at 4 h with diverse contents of thioether benzimidazole **7f**, and the generation of RNIs reduced and held constants after 4 h.

Excessive ROS and RNIs are in an unbalanced state with the antioxidant protection mechanism, leading to occurrence of oxidative stress and dysfunction of cells. Membrane lipid peroxidation is one of the manifestations of oxidative stress. Malondialdehyde (MDA) is an extremely significant product of membrane lipid peroxidation, so the determination of MDA can help to understand the degree of membrane lipid peroxidation and further understand the degree of oxidative damage [52,53]. The production of MDA in *C. tropicalis* treated by **7f** appeared a dose-dependent increase, which revealed the appearance of membrane lipid peroxidation and oxidative damage (Figure 8B).

Glutathione is a marker for assessing oxidative stress, and exists in both reduced form (GSH) and oxidative form (GSSG). The production of excess ROS in the organism interferences the equilibrium of the redox system and leads to the conversion of GSH into GSSG. This degree of GSH to GSSG transformation results in a reduction in GSH activity as an indicator of oxidative stress that can be quantified through the Ellman experiment [54]. The experimental result of *C. tropicalis* integrated with increasing amount of **7f** showed a continuous weakening of the GSH activity, and it was widely proved that the accumulation of ROS was advantageous to conquer the antioxidant defense system (Figure 8D). Moreover, the oxidative damage of the *C. tropicalis* undergoing treatment was assessed by Alamar blue (Resazurin) assay based on fluorescence spectra [55]. After cell was damaged, the Alamar blue dye turned into oxidation state (resazurin) from reduction state (resorufin) entering the cell, and the solution gradually changed from pink to blue (Figure 8E,F).

### 2.10. Measurement of Metabolic Activity

Alamar blue (Resazurin) assay was applied to assess the intracellular metabolic activity of the *C. tropicalis* during treatment and analyze the cell activity and cell proliferation of *C. tropicalis* strains [56]. Alamar blue does not exhibit fluorescence in the oxidized state, but in the reduced state, it occurs a reduction product by pink or red fluorescence. The Alamar blue dye entering the viable cells was reduced by metabolic intermediates (NADPH/NADP, FADH/FAD, FMNH/FMN and NADH/NAD) and cytochromes, released into the outside of cells, and transformed from the non-fluorescent indigo blue to the fluorescent pink. However, inactive or damaged cells possessed lower metabolic activity and lower corresponding signals. The result displayed in Figure 9 showed that the metabolic activity of *C. tropicalis* reduced upon treatment with thioether benzimidazole **7f**. At the increased concentrations of compound **7f**, metabolic activity was gradually decreased and finally metabolized inert. Thus, the decrease in metabolic activity clearly showed that the damage of cell membrane of *C. tropicalis* upon interacting with compound **7f** observably impeded the cellular respiration of *C. tropicalis*, which disorganized respiration and caused metabolic arrest and loss of cell viability.

### 2.11. Synergistic Effect of Chemical and Dynamic Antifungal Treatment for Hydroxyethyl Naphthalimide Antifungals

Based on the above, the prepared hydroxyethyl naphthalimides exhibited large inhibitory potentiality against the *C. tropicalis* strain through a synergistic effect of chemical and dynamic treatment, including DNA damage, membrane disruption, protein leakage, metabolic deactivation and oxidative damage (Figure 10).

## 3. Materials and Methods

### 3.1. Instruments and Chemicals

Melting points were recorded on X–6 melting point apparatus and were uncorrected. TLC analysis was done using pre-coated silica gel plates. The ^1^H NMR and ^13^C NMR spectra were recorded on a Bruker AVANCE III 600 MHz spectrometer using TMS as an internal standard. The chemical shifts (δ) were reported in parts per million (ppm), the coupling constants (*J*) were expressed in hertz (Hz) and signals were described as singlet (s), doublet (d), triplet (t) as well as multiplet (m). The high resolution mass spectra (HRMS) were recorded on Bruker Impact II (Bremen, Germany). The purity was measured by HITACHI primaide (Japan). All raw materials and solvents were commercially available and were used without further purification.

### 3.2. Synthesis of Hydroxyethyl Naphthalimides

#### 3.2.1. Synthesis of 6-Bromo-2-(2-hydroxyethyl)-1H-benzo[de]isoquinoline-1,3(2H)-dione (**2**)

A mixture of 4-bromo-1,8-naphthalic anhydride (3.0 g, 10.8 mmol), ethanolamine (1.0 mL, 11.9 mmol) and ethanol (150 mL) was stirred at 80 °C for 4 h. The mixture was cooled to room temperature and the solvent was removed. The solid was obtained without purification and used in the next step, yield: 86.7%; M.p. 203–204 °C.

#### 3.2.2. Synthesis of 6-(Dimethylamino)-2-(2-hydroxyethyl)-1H-benzo[de]isoquinoline-1,3 (2H)-dione (**3a**)

A mixture of **2** (300 mg, 0.94 mmol), dimethylamine (1 mL, 14.5 mmol), triethylamine (1.3 mL, 9.37 mmol) and 2-methoxyethanol (5 mL) was stirred at 100 °C for 6 h. The mixture was cooled to room temperature and the solvent was removed. The obtained solid was further purified by silica gel column chromatography (300–400 mesh) (Eluent: ethyl acetate/petroleum ether = 1/10~5, *V/V*) to produce yellow solid compound **3a** (124 mg); Yield: 46.4%; M.p. 203.5–204.5 °C; Purity: 99.9%. ^1^H NMR (600 MHz, DMSO-*d_6_*) δ 8.48 (d, *J* = 7.8 Hz, 1H, naphthalimide-*H*), 8.42 (d, *J* = 7.1 Hz, 1H, naphthalimide-*H*), 8.30 (d, *J* = 8.3 Hz, 1H, naphthalimide-*H*), 7.73 (m, 1H, naphthalimide-*H*), 7.18 (d, *J* = 7.9 Hz, 1H, naphthalimide-*H*), 4.80 (bs, 1H, O*H*), 4.13 (t, *J* = 6.5 Hz, 2H, C*H*_2_CH_2_OH), 3.60 (t, *J* = 5.4 Hz, 2H, C*H*_2_OH), 3.08 (s, 6H, C*H*_3_) ppm; ^13^C NMR (150 MHz, DMSO-*d*_6_) δ 164.21, 163.56 (*C*=O), 156.94, 132.62, 131.83, 130.91, 130.09, 125.41, 122.89, 113.44, 58.40, 44.85, 42.02, 34.78 ppm; HRMS (ESI) calcd. for C_16_H_16_N_2_O_3_ [M + H]^+^: 285.1234; found: 285.1234. The compounds are characterized in the Appendix A.

#### 3.2.3. Synthesis of 6-(Diethylamino)-2-(2-hydroxyethyl)-1H-benzo[de]isoquinoline-1,3(2H)-dione (**3b**)

Compound **3b** was prepared according to the procedure described for compound **3a**, starting from **2** (300 mg, 0.94 mmol), diethylamine (1 mL, 9.70 mmol), triethylamine (1.3 mL, 9.37 mmol) and 2-methoxyethanol (5 mL). The pure product **3b** was obtained as yellow solid (150 mg); Yield: 51.2%; M.p. 206.5–207.3 °C; Purity: 98.8%. ^1^H NMR (600 MHz, DMSO-*d_6_*) δ 8.70 (d, *J* = 8.5 Hz, 1H, naphthalimide-*H*), 8.43 (d, *J* = 7.2 Hz, 1H, naphthalimide-*H*), 8.32 (d, *J* = 8.2 Hz, 1H, naphthalimide-*H*), 7.73 (t, *J* = 7.9 Hz, 1H, naphthalimide-*H*), 7.27 (d, *J* = 8.2 Hz, 1H, naphthalimide-*H*), 4.79 (bs, 1H, O*H*), 4.13 (t, *J* = 6.5 Hz, 2H, C*H*_2_CH_2_OH), 3.60 (t, *J* = 5.4 Hz, 2H, C*H*_2_OH), 3.47 (q, *J* = 7.1 Hz, 4H, C*H*_2_CH_3_), 1.21 (t, *J* = 7.1 Hz, 6H, C*H*_3_) ppm; ^13^C NMR (150 MHz, DMSO-*d*_6_) δ 164.24, 163.58 (*C*=O), 157.13, 132.52, 131.87, 130.92, 130.09, 125.40, 125.19, 122.89, 114.62, 114.21, 59.76, 58.82, 47.56 (*C*H_2_), 12.25 (*C*H_3_) ppm; HRMS (ESI) calcd. for C_18_H_20_N_2_O_3_ [M + H]^+^: 313.1547; found: 313.1547.

#### 3.2.4. Synthesis of 2-(2-Hydroxyethyl)-6-((2-hydroxyethyl)(methyl)amino)-1H-benzo[de] isoquinoline-1,3(2H)-dione (**4a**)

Compound **4a** was prepared according to the procedure described for compound **3a**, starting from **2** (500 mg, 1.56 mmol), 2-methylaminoethanol (1.3 mL, 15.62 mmol), triethylamine (1.3 mL, 9.37 mmol) and 1,4-dioxane (5 mL). The pure product **4a** was obtained as yellow solid (311 mg); Yield: 63.5%; M.p. 207.5–208.1 °C; Purity: 99.1%. ^1^H NMR (600 MHz, DMSO-*d_6_*) δ 8.70 (d, *J* = 8.5 Hz, 1H, naphthalimide-*H*), 8.43 (d, *J* = 7.2 Hz, 1H, naphthalimide-*H*), 8.32 (d, *J* = 8.2 Hz, 1H, naphthalimide-*H*), 7.73 (t, *J* = 7.9 Hz, 1H, naphthalimide-*H*), 7.27 (d, *J* = 8.2 Hz, 1H, naphthalimide-*H*), 4.87 (bs, 1H, O*H*), 4.79 (bs, 1H, O*H*), 4.13 (t, *J* = 6.6 Hz, 2H, C*H*_2_CH_2_OH), 3.78 (t, *J* = 5.3 Hz, 2H, C*H*_2_OH), 3.61 (t, *J* = 5.6 Hz, 2H, C*H*_2_CH_2_OH), 3.43 (t, *J* = 5.8 Hz, 2H, C*H*_2_OH), 3.07 (s, 3H, C*H*_3_) ppm; ^13^C NMR (150 MHz, DMSO-*d*_6_) δ 164.24, 163.58 (*C*=O), 157.13, 132.52, 131.87, 130.92, 130.09, 125.40, 125.19, 122.89, 114.62, 114.21, 59.76, 58.82, 42.02, 40.87 ppm; HRMS (ESI) calcd. for C_17_H_18_N_2_O_4_ [M + H]^+^: 315.1339; found: 315.1336.

#### 3.2.5. Synthesis of 6-(Ethyl(2-hydroxyethyl)amino)-2-(2-hydroxyethyl)-1H-benzo[de] isoquinoline-1,3(2H)-dione (**4b**)

Compound **4b** was prepared according to the procedure described for compound **3a**, starting from **2** (500 mg, 1.56 mmol), 2-(ethylamino)ethanol (1.3 mL, 15.6 mmol), triethylamine (1.3 mL, 9.37 mmol) and 1,4-dioxane (5 mL). The pure product **4b** was obtained as yellow solid (267 mg); Yield: 52.2%; M.p. 234.5–235.3 °C; Purity: 99.3%. ^1^H NMR (600 MHz, DMSO-*d_6_*) δ 8.70 (d, *J* = 8.5 Hz, 1H, naphthalimide-*H*), 8.43 (d, *J* = 7.2 Hz, 1H, naphthalimide-*H*), 8.32 (d, *J* = 8.2 Hz, 1H, naphthalimide-*H*), 7.73 (t, *J* = 7.9 Hz, 1H, naphthalimide-*H*), 7.27 (d, *J* = 8.2 Hz, 1H, naphthalimide-*H*), 4.87 (bs, 1H, O*H*), 4.79 (bs, 1H, O*H*), 4.13 (t, *J* = 6.6 Hz, 2H, C*H*_2_CH_2_OH), 3.78 (t, *J* = 5.3 Hz, 2H, C*H*_2_OH), 3.61 (t, *J* = 5.6 Hz, 2H, C*H*_2_CH_2_OH), 3.50 (q, *J* = 7.0 Hz, 2H, C*H*_2_CH_3_), 3.43 (t, *J* = 5.8 Hz, 2H, C*H*_2_OH), 1.19 (t, *J* = 7.0 Hz, 3H, C*H*_3_) ppm; ^13^C NMR (150 MHz, DMSO-*d*_6_) δ 164.24, 163.58 (*C*=O), 157.13, 132.52, 131.87, 130.92, 130.09, 125.40, 125.19, 122.89, 114.62, 114.21, 59.76, 58.82, 42.02, 40.87, 11.97 ppm; HRMS (ESI) calcd. for C_18_H_20_N_2_O_4_ [M + H]^+^: 329.1496; found: 329.1493.

#### 3.2.6. Synthesis of 6-(Bis(2-hydroxyethyl)amino)-2-(2-hydroxyethyl)-1H-benzo[de] isoquinoline-1,3(2H)-dione (**4c**)

Compound **4c** was prepared according to the procedure described for compound **3a**, starting from **2** (500 mg, 1.56 mmol), diethanolamine (1.64 g, 15.6 mmol), triethylamine (1.3 mL, 9.37 mmol) and 1,4-dioxane (5 mL). The pure product **4c** was obtained as red solid (285 mg); Yield: 53.2%; M.p. 211.1–211.6 °C; Purity: 99.7%. ^1^H NMR (600 MHz, DMSO-*d_6_*) δ 8.44 (m, 2H, naphthalimide-*H*), 8.30 (d, *J* = 26.3 Hz, 1H, naphthalimide-*H*), 7.72 (m, 1H, naphthalimide-*H*), 7.18 (d, *J* = 25.0 Hz, 1H, naphthalimide-*H*), 4.80 (bs, 1H, O*H*), 4.13 (t, *J* = 6.6 Hz, 2H, C*H*_2_CH_2_OH), 3.61 (t, *J* = 5.3 Hz, 2H, C*H*_2_OH), 3.36 (t, *J* = 5.8 Hz, 2H, C*H*_2_CH_2_OH), 3.10 (t, *J* = 5.8 Hz, 2H, C*H*_2_CH_2_OH), 3.08 (m, 4H, C*H*_2_OH) ppm; ^13^C NMR (150 MHz, DMSO-*d*_6_) δ 164.24, 163.51 (*C*=O), 156.97, 132.64, 131.84, 130.85, 125.42, 124.74, 122.94, 114.01, 113.37, 42.02 ppm; HRMS (ESI) calcd. for C_18_H_20_N_2_O_5_ [M + H]^+^: 345.1445; found: 345.1445.

#### 3.2.7. Synthesis of (2-(2-Hydroxyethyl)-1,3-dioxo-2,3-dihydro-1H-benzo[de]isoquinolin-6-yl)proline (**5**)

Compound **5** was prepared according to the procedure described for compound **3a**, starting from **2** (723 mg, 2.56 mmol), L-proline (1.47 g, 12.8 mmol) and 2-methoxyethanol (10 mL). The pure product **5** was obtained as yellow solid (455 mg); Yield: 57.1%; M.p. 198.6–199.2 °C; Purity: 99.4%. ^1^H NMR (600 MHz, CD_3_OD) δ 9.48 (d, *J* = 8.8 Hz, 1H, naphthalimide-*H*), 9.20 (d, *J* = 7.2 Hz, 1H, naphthalimide-*H*), 9.00 (d, *J* = 8.6 Hz, 1H, naphthalimide-*H*), 8.39 (t, *J* = 7.9 Hz, 1H, naphthalimide-*H*), 7.63 (d, *J* = 8.9 Hz, 1H, naphthalimide-*H*), 4.92 (t, *J* = 6.8 Hz, 2H, C*H*_2_CH_2_OH), 4.82 (bs, 1H, CH_2_CH_2_O*H*), 4.39 (t, *J* = 6.8 Hz, 2H, CH_2_C*H*_2_OH), 4.06 (d, *J* = 7.2 Hz, 1H, C*H*COOH), 3.81 (m, 2H, pyrrolidine-*H*), 3.21 (m, 2H, pyrrolidine-*H*), 2.85 (m, 2H, pyrrolidine-*H*) ppm; HRMS (ESI) calcd. for C_19_H_18_N_2_O_5_ [M + Na]^+^, 377.1108; found, 377.1108.

#### 3.2.8. Synthesis of 2-(2-Hydroxyethyl)-6-methoxy-1H-benzo[de]isoquinoline-1,3(2H)-dione (**6a**)

Compound **6a** was prepared according to the procedure described for compound **3a**, starting from **2** (500 mg, 1.56 mmol), potassium carbonate (170 mg, 1.23 mmol) and methanol (20 mL). The pure product **6a** was obtained as yellow solid (276 mg); Yield: 65.4%; M.p. 189.3–189.9 °C; Purity: 99.3%. ^1^H NMR (600 MHz, CDCl_3_) δ 8.60 (d, *J* = 7.2 Hz, 1H, naphthalimide-*H*), 8.57 (d, *J* = 5.6 Hz, 1H, naphthalimide-*H*), 8.56 (d, *J* = 5.6 Hz, 1H, naphthalimide-*H*), 7.70 (t, *J* = 7.8 Hz, 1H, naphthalimide-*H*), 7.05 (d, *J* = 8.3 Hz, 1H, naphthalimide-*H*), 4.45 (t, *J* = 5.2 Hz, 2H, C*H*_2_CH_2_OH), 4.14 (s, 3H, C*H*_3_), 3.98 (t, *J* = 5.2 Hz, 2H, C*H*_2_OH) ppm; ^13^C NMR (150 MHz, CDCl_3_) δ 160.64, 160.11 (*C*=O), 156.35, 129.09, 127.09, 124.70, 124.22, 121.22, 118.75, 117.36, 110.01, 51.49, 38.01 ppm; HRMS (ESI) calcd. for C_15_H_13_NO_4_ [M + H]^+^, 272.0917; found, 272.0917.

#### 3.2.9. Synthesis of 6-Ethoxy-2-(2-hydroxyethyl)-1H-benzo[de]isoquinoline-1,3(2H)-dione (**6b**)

Compound **6b** was prepared according to the procedure described for compound **3a**, starting from **2** (500 mg, 1.56 mmol), potassium carbonate (170 mg, 1.23 mmol) and ethanol (20 mL). The pure product **6b** was obtained as yellow solid (268 mg); Yield: 60.3%; M.p. 192.3–192.6 °C; Purity: 99.3%. ^1^H NMR (600 MHz, CDCl_3_) δ 8.60 (d, *J* = 7.2 Hz, 1H, naphthalimide-*H*), 8.57 (d, *J* = 5.6 Hz, 1H, naphthalimide-*H*), 8.56 (d, *J* = 5.6 Hz, 1H, naphthalimide-*H*), 7.70 (t, *J* = 7.8 Hz, 1H, naphthalimide-*H*), 7.05 (d, *J* = 8.3 Hz, 1H, naphthalimide-*H*), 4.45 (t, *J* = 5.2 Hz, 2H, C*H*_2_CH_2_OH), 4.62 (m, 2H, C*H*_2_CH_3_), 3.98 (t, *J* = 5.2 Hz, 2H, C*H*_2_OH), 1.55 (t, *J* = 7.2 Hz, 3H, CH_2_C*H*_3_) ppm; ^13^C NMR (150 MHz, CDCl_3_) δ 164.64, 163.11 (*C*=O), 156.35, 129.09, 127.09, 124.70, 124.22, 121.22, 118.75, 117.36, 110.01, 51.49, 45.01, 23.34, 11.23 ppm; HRMS (ESI) calcd. for C_16_H_15_NO_4_ [M + H]^+^, 286.1074; found, 286.1074.

#### 3.2.10. Synthesis of 2-(2-Hydroxyethyl)-6-(2-methoxyethoxy)-1H-benzo[de]isoquinoline-1,3(2H)-dione (**6c**)

Compound **6c** was prepared according to the procedure described for compound **3a**, starting from **2** (500 mg, 1.56 mmol), potassium carbonate (170 mg, 1.23 mmol) and 2-methoxyethanol (20 mL). The pure product **6c** was obtained as yellow solid (272 mg); Yield: 55.4%; M.p. 215.4–215.9 °C; Purity: 99.7%. ^1^H NMR (600 MHz, CDCl_3_) δ 8.61 (d, *J* = 7.3 Hz, 1H, naphthalimide-*H*), 8.59 (d, *J* = 7.3 Hz, 1H, naphthalimide-*H*), 8.53 (d, *J* = 8.3 Hz, 1H, naphthalimide-*H*), 7.70 (t, *J* = 7.3 Hz, 1H, naphthalimide-*H*), 7.04 (d, *J* = 8.3 Hz, 1H, naphthalimide-*H*), 4.45 (bs, 2H, C*H*_2_CH_2_OH), 4.42 (bs, 2H, C*H*_2_CH_2_OCH_3_), 3.97 (bs, 2H, C*H*_2_OH), 3.94 (bs, 2H, CH_2_C*H*_2_OCH_3_), 3.52 (s, 3H, C*H*_3_) ppm; ^13^C NMR (150 MHz, CDCl_3_) δ 165.33, 164.77 (*C*=O), 160.27, 133.69, 131.85, 129.11, 125.94, 123.53, 122.07, 114.89, 106.06, 70.66, 68.47, 61.99, 59.35, 42.75 ppm; HRMS (ESI) calcd. for C_17_H_17_NO_5_ [M + H]^+^, 316.1180; found, 316.1179.

#### 3.2.11. Synthesis of 2-(2-Hydroxyethyl)-6-((1-methyl-1H-imidazol-2-yl)thio)-1H-benzo[de] isoquinoline-1,3(2H)-dione (**7a**)

Compound **7a** was prepared according to the procedure described for compound **3a**, starting from **2** (500 mg, 1.56 mmol), 2-mercapto-1-methylimidazole (214 mg, 1.87 mmol), potassium carbonate (216 mg, 1.56 mmol) and *N*,*N*-dimethylformamide (7 mL). The pure product **7a** was obtained as yellow solid (356 mg); Yield: 64.7%; M.p. >250 °C; Purity: 99.9%. ^1^H NMR (600 MHz, DMSO-*d*_6_) δ 8.67 (d, *J* = 8.4 Hz, 1H, naphthalimide-*H*), 8.55 (d, *J* = 7.3 Hz, 1H, naphthalimide-*H*), 8.31 (d, *J* = 7.9 Hz, 1H, naphthalimide-*H*), 7.96 (t, *J* = 7.9 Hz, 1H, naphthalimide-*H*), 7.62 (bs, 1H, imidazole-*H*), 7.26 (bs, 1H, imidazole-*H*), 7.01 (d, *J* = 7.9 Hz, 1H, naphthalimide-*H*), 4.79 (bs, 1H, O*H*), 4.13 (t, *J* = 6.5 Hz, 2H, C*H*_2_CH_2_OH), 3.65 (s, 3H C*H*_3_), 3.61 (t, *J* = 6.4 Hz, 2H, C*H*_2_OH) ppm; ^13^C NMR (150 MHz, DMSO-*d*_6_) δ 163.67, 163.53 (*C*=O), 142.14, 133.60, 131.70, 131.05, 131.00, 129.93, 128.67, 128.34, 128.24, 126.65, 125.20, 123.43, 120.82, 58.25, 42.34, 34.04 ppm; HRMS (ESI) calcd. for C_18_H_15_N_3_O_3_S [M + H]^+^, 354.0907; found, 354.0907.

#### 3.2.12. Synthesis of 2-(2-Hydroxyethyl)-6-((1-methyl-1H-tetrazol-5-yl)thio)-1H-benzo[de] isoquinoline-1,3(2H)-dione (**7b**)

Compound **7b** was prepared according to the procedure described for compound **3a**, starting from **2** (500 mg, 1.56 mmol), 1-methyl-1H-tetrazole-5-thiol (217 mg, 1.87 mmol), potassium carbonate (170 mg, 1.23 mmol) and *N*,*N*-dimethylformamide (10 mL). The pure product **7b** was obtained as yellow solid (278 mg); Yield: 50.3%; M.p. >250 °C; Purity: 99.9%. ^1^H NMR (600 MHz, DMSO-*d*_6_) δ 8.66 (d, *J* = 8.4 Hz, 1H, naphthalimide-*H*), 8.53 (d, *J* = 7.2 Hz, 1H, naphthalimide-*H*), 8.40 (d, *J* = 7.7 Hz, 1H, naphthalimide-*H*), 7.96 (t, *J* = 7.9 Hz, 1H, naphthalimide-*H*), 7.90 (d, *J* = 7.7 Hz, 1H, naphthalimide-*H*), 4.80 (bs, 1H, O*H*), 4.14 (t, *J* = 6.3 Hz, 2H, C*H*_2_CH_2_OH), 4.12 (s, 3H, C*H*_3_), 3.63 (t, *J* = 6.3 Hz, 2H, C*H*_2_OH) ppm; ^13^C NMR (150 MHz, DMSO-*d*_6_) δ 163.53, 163.34 (*C*=O), 151.25, 134.11, 132.48, 131.74, 131.03, 130.97, 130.73, 128.54, 123.77, 123.52, 58.23, 42.46, 34.92 ppm; HRMS (ESI) calcd. for C_16_H_13_N_5_O_3_S [M + H]^+^, 356.0812; found, 356.0810.

#### 3.2.13. Synthesis of 6-((1H-1,2,4-Triazol-5-yl)thio)-2-(2-hydroxyethyl)-1H-benzo[de] isoquinoline-1,3(2H)-dione (**7c**)

Compound **7c** was prepared according to the procedure described for compound **3a**, starting from **2** (500 mg, 1.56 mmol), 1H-1,2,4-triazole-3-thiol (190 mg, 1.87 mmol), potassium carbonate (170 mg, 1.23 mmol) and *N*,*N*-dimethylformamide (10 mL). The pure product **7c** was obtained as yellow solid (246 mg); Yield: 46.4%; M.p. >250 °C; Purity: 99.9%. ^1^H NMR (600 MHz, DMSO-*d*_6_) δ 14.59 (s, 1H, N*H*), 8.80 (s, 1H, triazole-*H*), 8.62 (d, *J* = 8.0 Hz, 1H, naphthalimide-*H*), 8.54 (d, *J* = 6.4 Hz, 1H, naphthalimide-*H*), 8.36 (d, *J* = 6.6 Hz, 1H, naphthalimide-*H*), 7.93 (t, *J* = 8.2 Hz, 1H, naphthalimide-*H*), 7.66 (d, *J* = 6.5 Hz, 1H, naphthalimide-*H*), 4.80 (bs, 1H, O*H*), 4.13 (t, *J* = 6.8 Hz, 2H, C*H*_2_CH_2_OH), 3.63 (t, *J* = 8.1 Hz, 2H, C*H*_2_OH) ppm; ^13^C NMR (150 MHz, DMSO-*d*_6_) δ 163.67, 163.53 (*C*=O), 146.55, 131.55, 130.74, 130.55, 128.72, 128.31, 128.23, 123.39, 58.26, 42.38 ppm; HRMS (ESI) calcd. for C_16_H_12_N_4_O_3_S [M + H]^+^, 341.0703; found, 341.0700.

#### 3.2.14. Synthesis of 2-(2-Hydroxyethyl)-6-((5-methyl-1,3,4-thiadiazol-2-yl)thio)-1H-benzo [de]isoquinoline-1,3(2H)- dione (**7d**)

Compound **7d** was prepared according to the procedure described for compound **3a**, starting from **2** (500 mg, 1.56 mmol), 5-methyl-1,3,4-thiadiazole-2-thiol (247 mg, 1.87 mmol), potassium carbonate (170 mg, 1.23 mmol) and *N*,*N*-dimethylformamide (10 mL). The pure product **7d** was obtained as yellow solid (326 mg); Yield: 56.4%; M.p. >250 °C; Purity: 99.9%. ^1^H NMR (600 MHz, CDCl_3_) δ 8.74 (d, *J* = 8.5 Hz, 1H, naphthalimide-*H*), 8.67 (d, *J* = 7.2 Hz, 1H, naphthalimide-*H*), 8.56 (d, *J* = 7.6 Hz, 1H, naphthalimide-*H*), 8.07 (d, *J* = 7.6 Hz, 1H, naphthalimide-*H*), 7.85 (t, *J* = 7.9 Hz, 1H, naphthalimide-*H*), 4.45 (t, *J* = 5.3 Hz, 2H, C*H*_2_CH_2_OH), 3.98 (t, *J* = 5.3 Hz, 2H, C*H*_2_OH), 2.70 (s, 3H, C*H*_3_) ppm; ^13^C NMR (150 MHz, CDCl_3_) δ 167.70, 164.40, 164.19, 163.33 (*C*=O), 136.76, 133.14, 132.27, 131.62, 131.42, 130.95, 128.97, 128.32, 123.90, 123.18, 61.50, 42.88, 15.80 ppm; HRMS (ESI) calcd. for C_17_H_13_N_3_O_3_S_2_ [M + H]^+^, 372.0471; found, 372.0470.

#### 3.2.15. Synthesis of 6-(Benzo[d]thiazol-2-ylthio)-2-(2-hydroxyethyl)-1H-benzo[de] isoquinoline-1,3(2H)-dione (**7e**)

Compound **7e** was prepared according to the procedure described for compound **3a**, starting from **2** (300 mg, 0.94 mmol), 2-mercaptobenzothiazole (187 mg, 1.12 mmol), potassium carbonate (130 mg, 0.94 mmol) and *N*,*N*-dimethylformamide (10 mL). The pure product **7e** was obtained as yellow solid (241 mg); Yield: 63.2%; M.p. >250 °C; Purity: 98.8%. ^1^H NMR (600 MHz, DMSO-*d*_6_) δ 8.69 (d, *J* = 14.2 Hz, 1H, naphthalimide-*H*), 8.53 (m, 2H, benzothiazole-*H*), 8.37 (d, *J* = 15.1 Hz, 1H, naphthalimide-*H*), 7.94 (d, *J* = 15.1 Hz, 1H, naphthalimide-*H*), 7.87 (m, 2H, benzothiazole-*H*), 7.45 (d, *J* = 7.2 Hz, 1H, naphthalimide-*H*), 7.34 (d, *J* = 7.1 Hz, 1H, naphthalimide-*H*), 4.84 (bs, 1H, O*H*), 4.16 (t, *J* = 6.4 Hz, 2H, C*H*_2_CH_2_OH), 3.67 (t, *J* = 6.2 Hz, 2H, C*H*_2_OH) ppm; ^13^C NMR (150 MHz, DMSO-*d*_6_) δ 165.97, 163.51, 163.33 (*C*=O), 153.41, 136.12, 135.68, 134.33, 132.23, 131.84, 131.41, 130.76, 129.34, 128.75, 127.11, 125.47, 123.67, 122.33, 122.23, 58.25, 42.55 ppm; HRMS (ESI) calcd. for C_21_H_14_N_2_O_3_S_2_ [M + H]^+^, 407.0519; found, 407.0514.

#### 3.2.16. Synthesis of 6-((1H-Benzo[d]imidazol-2-yl)thio)-2-(2-hydroxyethyl)-1H-benzo [de]isoquinoline-1,3(2H)-dione (**7f**)

Compound **7f** was prepared according to the procedure described for compound **3a**, starting from **2** (500 mg, 1.56 mmol), 2-mercaptobenzimidazole (281 mg, 1.87 mmol), potassium carbonate (170 mg, 1.23 mmol) and *N*,*N*-dimethylformamide (10 mL). The pure product **7f** was obtained as yellow solid (275 mg); Yield: 45.3%; M.p. >250 °C; Purity: 99.9%. ^1^H NMR (600 MHz, DMSO-*d*_6_) δ 13.04 (s, 1H, N*H*), 8.66 (d, *J* = 8.4 Hz, 1H, naphthalimide-*H*), 8.55 (d, *J* = 7.2 Hz, 1H, naphthalimide-*H*), 8.42 (d, *J* = 7.7 Hz, 1H, naphthalimide-*H*), 7.94 (t, *J* = 7.9 Hz, 1H, naphthalimide-*H*), 7.86 (d, *J* = 7.7 Hz, 1H, naphthalimide-*H*), 7.51 (m, 2H, benzimidazole-*H*), 7.21 (m, 2H, benzimidazole-*H*), 4.83 (bs, 1H, O*H*), 4.16 (t, *J* = 6.4 Hz, 2H, C*H*_2_CH_2_OH), 3.64 (t, *J* = 6.6 Hz, 2H, C*H*_2_OH) ppm; ^13^C NMR (150 MHz, DMSO-*d*_6_) δ 163.64, 163.49 (*C*=O), 144.99, 137.65, 131.64, 131.17, 130.92, 130.72, 128.57, 128.52, 123.44, 122.69, 58.26, 42.40 ppm; HRMS (ESI) calcd. for C_21_H_15_N_3_O_3_S [M + H]^+^, 390.0907; found, 390.0906.

#### 3.2.17. Synthesis of 2-(2-Hydroxyethyl)-6-(pyrimidin-2-ylthio)-1H-Benzo[de]isoquinoline-1,3(2H)-dione (**8a**)

Compound **8a** was prepared according to the procedure described for compound **3a**, starting from **2** (300 mg, 0.94 mmol), pyrimidine-2-thiol (126 mg, 1.12 mmol), potassium carbonate (130 mg, 0.94 mmol) and *N*,*N*-dimethylformamide (10 mL). The pure product **8a** was obtained as yellow solid (143 mg); Yield: 43.3%; M.p. >250 °C; Purity: 99.9%. ^1^H NMR (600 MHz, CDCl_3_) δ 8.67 (d, *J* = 8.4 Hz, 1H, naphthalimide-*H*), 8.64 (d, *J* = 7.2 Hz, 1H, naphthalimide-*H*), 8.62 (d, *J* = 7.6 Hz, 1H, naphthalimide-*H*), 8.41 (d, *J* = 4.8 Hz, 2H, pyrimidine-*H*), 8.16 (d, *J* = 7.5 Hz, 1H, naphthalimide-*H*), 7.75 (t, *J* = 7.9 Hz, 1H, naphthalimide-*H*), 7.00 (t, *J* = 4.8 Hz, 1H, pyrimidine-*H*), 4.47 (t, *J* = 5.3 Hz, 2H, C*H*_2_CH_2_OH), 3.99 (t, *J* = 5.3 Hz, 2H, C*H*_2_OH) ppm; ^13^C NMR (150 MHz, CDCl_3_) δ 171.44, 164.74, 164.57 (*C*=O), 135.69, 135.42, 133.33, 132.59, 131.87, 130.96, 128.90, 127.71, 123.87, 123.00, 117.62, 61.62, 42.84 ppm; HRMS (ESI) calcd. for C_18_H_13_N_3_O_3_S [M + H]^+^, 352.0750; found, 352.0755.

#### 3.2.18. Synthesis of 2-(2-Hydroxyethyl)-6-((4-methylpyrimidin-2-yl)thio)-1H-benzo[de] isoquinoline-1,3(2H)-dione (**8b**)

Compound **8b** was prepared according to the procedure described for compound **3a**, starting from **2** (300 mg, 0.94 mmol), 4-methylpyrimidine-2-thiol (142 mg, 1.12 mmol), potassium carbonate (130 mg, 0.94 mmol) and *N*,*N*-dimethylformamide (10 mL). The pure product **8b** was obtained as yellow solid (190 mg); Yield: 55.3%; M.p. >250 °C; Purity: 99.5%. ^1^H NMR (600 MHz, CDCl_3_) δ 8.65 (d, *J* = 8.4 Hz, 1H, naphthalimide-*H*), 8.62 (d, *J* = 7.2 Hz, 1H, naphthalimide-*H*), 8.59 (d, *J* = 7.5 Hz, 1H, naphthalimide-*H*), 8.19 (d, *J* = 5.0 Hz, 1H, naphthalimide-*H*), 8.14 (d, *J* = 7.5 Hz, 1H, pyrimidine-*H*), 7.73 (t, *J* = 7.9 Hz, 1H, naphthalimide-*H*), 6.85 (d, *J* = 5.0 Hz, 1H, pyrimidine-*H*), 4.46 (t, *J* = 5.3 Hz, 2H, C*H*_2_CH_2_OH), 4.00 (t, *J* = 5.3 Hz, 2H, C*H*_2_OH), 2.39 (s, 3H, C*H*_3_) ppm; ^13^C NMR (150 MHz, CDCl_3_) δ 170.64, 168.35, 164.75, 164.60, 157.22, 135.86, 135.49, 133.24, 132.65, 131.78, 130.87, 128.80, 127.53, 123.59, 122.90, 117.36, 61.56, 42.83, 24.00 ppm; HRMS (ESI) calcd. for C_19_H_15_N_3_O_3_S [M + H]^+^, 366.0907; found, 366.0916.

#### 3.2.19. Synthesis of 6-((4,6-Dimethylpyrimidin-2-yl)thio)-2-(2-hydroxyethyl)-1H-benzo[de] isoquinoline-1,3(2H)-dione (**8c**)

Compound **8c** was prepared according to the procedure described for compound **3a**, starting from **2** (300 mg, 0.94 mmol), 4,6-dimethylpyrimidine-2-thiol (157 mg, 1.12 mmol), potassium carbonate (130 mg, 0.94 mmol) and *N*,*N*-dimethylformamide (10 mL). The pure product **8c** was obtained as yellow solid (183 mg); Yield: 51.3%; M.p. >250 °C; Purity: 99.5%. ^1^H NMR (600 MHz, CDCl_3_) δ 8.65 (d, *J* = 8.5 Hz, 1H, naphthalimide-*H*), 8.61 (d, *J* = 7.2 Hz, 1H, naphthalimide-*H*), 8.57 (d, *J* = 7.6 Hz, 1H, naphthalimide-*H*), 8.13 (d, *J* = 7.6 Hz, 1H, naphthalimide-*H*), 7.72 (t, *J* = 7.9 Hz, 1H, naphthalimide-*H*), 6.73 (s, 1H, pyrimidine-*H*), 4.47 (t, *J* = 5.3 Hz, 2H, C*H*_2_CH_2_OH), 4.01 (t, *J* = 5.3 Hz, 2H, C*H*_2_OH), 2.26 (s, 6H, C*H*_3_) ppm; ^13^C NMR (150 MHz, CDCl_3_) δ 169.74, 167.76, 164.84, 164.71, 136.52, 135.17, 133.12, 132.72, 131.69, 130.78, 128.72, 127.29, 123.20, 122.79, 116.93, 61.56, 42.84, 23.72 ppm; HRMS (ESI) calcd. for C_20_H_17_N_3_O_3_S [M + H]^+^, 380.1063; found, 380.1065.

#### 3.2.20. Synthesis of 6-((4-Hydroxy-6-methylpyrimidin-2-yl)thio)-2-(2-hydroxyethyl)-1H-benzo[de]isoquinoline-1,3(2H)-dione (**8d**)

Compound **8d** was prepared according to the procedure described for compound **3a**, starting from **2** (300 mg, 0.94 mmol), 2-mercapto-6-methylpyrimidin-4-ol (160 mg, 1.12 mmol), potassium carbonate (130 mg, 0.94 mmol) and *N*,*N*-dimethylformamide (10 mL). The pure product **8d** was obtained as yellow solid (166 mg); Yield: 46.3%; M.p. >250 °C; Purity: 99.5%. ^1^H NMR (600 MHz, DMSO-*d_6_*) δ 12.94 (s, 1H, pyrimidine-O*H*), 8.57 (d, *J* = 7.7 Hz, 1H, naphthalimide-*H*), 8.51 (d, *J* = 7.2 Hz, 1H, naphthalimide-*H*), 8.11 (d, *J* = 8.4 Hz, 1H, naphthalimide-*H*), 7.85 (d, *J* = 8.0 Hz, 1H, naphthalimide-*H*), 7.82 (d, *J* = 7.7 Hz, 1H, naphthalimide-*H*), 6.07 (s, 1H, pyrimidine-*H*), 4.84 (bs, 1H, CH_2_CH_2_O*H*), 4.18 (t, *J* = 6.4 Hz, 2H, C*H*_2_CH_2_OH), 3.65 (t, *J* = 6.2 Hz, 2H, C*H*_2_OH), 2.26 (s, 3H, C*H*_3_) ppm; ^13^C NMR (150 MHz, DMSO-*d*_6_) δ 177.80, 163.85, 163.52, 161.12, 153.96, 142.08, 131.25, 131.16, 129.38, 128.87, 128.64, 128.39, 123.44, 123.08, 103.94, 58.33, 42.46, 18.79 ppm; HRMS (ESI) calcd. for C_19_H_15_N_3_O_4_S [M + H]^+^, 382.0856; found, 382.0853.

### 3.3. Biological Assay

#### 3.3.1. Antifungal Assay

The newly synthesized compounds **2**, **3a**–**b**, **4a**–**c**, **5**, **6a**–**c**, **7a**–**f** and **8a**–**d** were evaluated for their antifungal activities against *Candida albicans* (*C. albicans*), *Candida albicans* ATCC 90023 (*C. albicans* 90023), *Candida tropicalis* (*C. tropicalis*), *Aspergillus fumigatus* (*A. fumigatus*), *Candida parapsilosis* ATCC 22019 (*C. parapsilosis* 22019). A spore suspension in sterile distilled water was prepared from one day old culture of the fungi growing on Sabouraud Agar (SA) media. The final spore concentration was 1–5 × 10^3^ spore mL^−1^. The tested compounds and reference fluconazole were dissolved in DMSO to prepare the stock solutions, and diluted in sterile RPM1 1640 medium (Neuronbc Laboraton Technology C1., Ltd., Beijing, China) to get eleven wanted concentrations of each tested compound. These dilutions were inoculated and incubated at 37 °C for 24 h.

#### 3.3.2. UV Absorption Spectra of Fluorophores with DNA

UV spectra were recorded at room temperature on a TU-2450 spectrophotometer (Puxi Analytic Instrument Ltd. of Beijing, China) equipped with 1.0 cm quartz cells. The stock solutions of fluorophores were prepared in DMSO. Tris-HCl buffer solution (pH = 7.4) was prepared by mixing and diluting Tris (tris(hydroxymethyl)aminomethane) solution with HCl solution. Tris and HCl were analytical purity. Sample masses were weighed on a microbalance with a resolution of 0.1 mg. All other chemicals and solvents were commercially available, and were used without further purification.

#### 3.3.3. Competitive Reaction of Compound **7f** and AO or DAPI with DNA

The fluorescence emission spectra of compound **7f** with AO-DNA and DAPI-DNA were recorded. The stock solution of compound **7f** was prepared in DMSO, and acridine orange (AO) and 4′,6-diamidino-2-phenylindole (DAPI) were prepared in distilled water. Tris-HCl buffer solution (pH = 7.4) was prepared by mixing and diluting Tris (tris(hydroxymethyl)aminomethane) solution with HCl solution. Tris and HCl were analytical purity. All other chemicals and solvents were commercially available, and were used without further purification.

#### 3.3.4. Measurement of Intracellular ROS Production

Intracellular ROS was measured using standard 2,7-dichlorofluoroscein diacetate (DCFH-DA) assay [57,58]. Then, 10^6^ CFU/mL of *Candida tropicalis* was treated with increasing concentrations of compound **7f** for 6 h at 37 °C and 200 rpm. Following treatment, both control and treated cells were washed with PBS and incubated with 100 μM DCFH-DA probe for 30 min in dark at 37 °C. The green fluorescence originating from the oxidative cleavage of DCFH-DA to DCF was measured in a microplate reader with an excitation wavelength of 485 nm and emission wavelength of 528 nm. The increase in intracellular ROS production in cells treated with compound **7f** in comparison to control cells was plotted.

#### 3.3.5. Measurement of RNIs by Griess’s Reaction

RNIs was measured using a spectrophotometric analysis of the total nitrite performed by using Griess’s reagent [59,60]. The *Candida tropicalis* suspension (100 μL) were incubated with 100 μL of compound **7f** (2 × MIC, 8 × MIC) at different times (1, 2, 3, 4, 5 and 6 h) at 37 °C. Then, 50 μL of 2% sulfanilamide in 5% (*v/v*) HCl and 50 μL of 0.1% N-(1-naphthyl)ethylenediamine dihydrochloride aqueous solution were added. The formation of the azo dye was measured 15 min later by spectrophotometry at 540 nm. The OD was directly proportional to the nitrite content of the standard solution. Results were expressed respect to control without compound **7f**.

#### 3.3.6. Measurement of MDA

Malondialdehyde (MDA) content of cell-free extract was determined using microplate reader. Briefly, cell-free extract was mixed with TBA/TCA/HCl (15%, 0.37%) at a reagent/sample ratio of 2:1 (*v/v*), placed in a boiling water bath for 15 min, cooled to room temperature, and centrifuged at 1000× *g* for 10 min at room temperature. The absorbance of the solution was read at 535 nm against the blank using microplate reader.

#### 3.3.7. Measurement of Intracellular Glutathione (GSH) Activity

The activity of intracellular GSH was determined using standard Ellman’s assay [61]. Then, 10^6^ CFU/mL of *Candida tropicalis* was treated with increasing concentrations of compound **7f** for 6 h at 37 °C and 200 rpm. Following treatment, both control and treated cells were centrifuged at 5000 rpm for 5 min, washed with PBS, and lysed. The lysed cells were further centrifuged, and the clear supernatant was collected. The supernatant was mixed with 50 mM Tris-HCl and 100 mM 5,5-dithiobis(2-nitrobenzoic acid) (DTNB) and incubated for 30 min in dark at 37 °C. The absorbance of the resulting solution was measured at 412 nm using microplate reader.

#### 3.3.8. Measurement of Alamar Blue Assay

Following 48 h of *C. tropicalis* growth, the media were replaced with fresh media containing increasing concentrations of compound **7f** (MIC, 2 × MIC, 4 × MIC, 6 × MIC and 8 × MIC). The strain was treated with compound **7f** for 24 h at 37 °C in a moist environment under static conditions. Following 24 h of treatment, the media were removed from the wells, and the strain was washed twice with PBS carefully to remove planktonic cells. Then, 100 μL of LB broth containing 10 μL of 5 μg/mL resazurin was added to the wells, and the plate was incubated for 45 min at 37 °C. Then, took photos for these wells, and fluorescence was measured at 571 nm excitation and 590 nm emission.

#### 3.3.9. Drug Resistance Development Assay

The strain of *C. tropicalis* was exposed to sub-MICs of compound **7f** for sustained passages, which determined every 24 h after propagation of *C. tropicalis* cultures and then the MIC of **7f** were determined against each passage of the strain. To make comparative analysis, fluconazole was used as the control experiment. The experiment was sustained for 16 passages.

#### 3.3.10. Hemolysis Assay

After washing and resuspending in PBS, 2% of human red blood cell was added to a 96-well plate with 100 μL per well. Then, the same volume of compound **7f** in various concentrations was added. 0.5% Triton X-100 (*v:v*) and PBS were used as positive control and negative control, respectively. After co-incubation at room temperature for one hour, the plate was centrifuged at 1500 rpm for 10 min. The absorbance of 100 μL of the supernatant was measured at 450 nm. The experiments were performed in triplicate, and the hemolysis percentage was calculated as follows: Hemolysis (%) = (A**_7f_** − A_PBS_)/(A_Triton_ − A_PBS_) × 100%.

#### 3.3.11. In Vitro Cytotoxicity

The cytotoxicity assays were determined with LO2 cells under normal training conditions. LO2 cells were inoculated into a sterile 96-well plates with a density of 4 × 10^−4^ cells·mL^−1^. Compound **7f** was put in DMSO and diluted with culture media. After 24 h, **7f** were put in the cultured LO2 cells for 24 h. Cell viability was determined by measuring the absorbance of the 3-[4,5-dimethylthiazol-2-yl]-2,5-diphenpyltetra-zolium bromide (MTT) assay at 570 nm. Each test was conducted in triplicate.

#### 3.3.12. Membrane Depolarization Assay

*Candida tropicalis* strain in their mid log phase (OD_600_ = 0.4–0.5) were washed with a buffer solution (5 mM HEPES buffer, 5 mM glucose, pH 7.2) and redispersed in the same buffer to an OD_600_ of 0.1. The redispersed cells were then incubated with 0.4 μM of 3,3′-dipropylthiadicarbocyanine iodide (diSC35) dye for 1 h at 37 °C, following which 100 mM KCl was added to the suspensions. After incubation with dye, the *Candida tropicalis* strain was treated with compound **7f** at MIC concentration, and the fluorescence of the treated cells was monitored periodically over a period of 1 h in fluorescence photometer set to an excitation wavelength of 622 nm and emission wavelength of 670 nm. Increase in fluorescence with time indicated membrane depolarization.

#### 3.3.13. Protein Leakage Assay

*Candida tropicalis* (10^6^ CFU/mL) was treated with increasing concentrations of compound **7f** for 6 h at 37 °C and 200 rpm. Following treatment, the cell was pelleted down at 5000 rpm for 5 min, and the cell-free supernatant was collected. The concentration of leaked proteins in the supernatant was measured using standard Bradford assay.

#### 3.3.14. Measurement of Metabolic Activity

The metabolic activity of *C. tropicalis* was measured using Alamar blue assay which is based on the ability of cells to convert a purple nonfluorescent dye resazurin to its pink fluorescent reduced form resofurin. Then, 10^6^ CFU/mL of *C. tropicalis* was treated with increasing concentrations of compound **7f** for 6 h at 37 °C and 200 rpm. Both control and treated cells were incubated with 25 μL of 50 μg/mL resazurin solution for 1 h at 37 °C. The metabolic conversion of resazurin to pink colored resofurin was quantified spectrophotometrically by measuring absorbance at 571 nm.

#### 3.3.15. Molecular Docking

The structure of cytochrome P450 reductase (CPR) employed in the docking calculations was obtained using RCSB Protein Data Bank (PDB ID: 6T1U). The structures of compound **7f** were drawn with ChemDraw 19.0. Docking analyses were performed with the Sybyl-X 2.0 and pymol program. The gird size was set to be 45 × 45 × 45 and the grid point spacing was set at default value 0.375 Å. The Lamarkian genetic algorithm (LGA) was applied for the conformational search.

## 4. Conclusions

In conclusion, a desirable family of hydroxyethyl naphthalimides with synergistic chemical and dynamic antifungal treatment were favourably discovered. These prepared compounds showed significant antifungal potency towards some tested fungi including *A. fumigatus*, *C. tropicalis* and *C. parapsilosis* 22019. Especially, thioether benzimidazole **7f** with excellent DNA binding ability gave better anti-*C. tropicalis* efficacy than fluconazole. Moreover, **7f** presented low cytotoxicity, safe hemolysis level and no obvious resistance. The strong lipase affinity of **7f** facilitated its permeation into cell membrane to cause membrane dysfunction. The studies of biological mechanisms directed by ROS and RNIs indicated prominent enhancement of intracellular oxidative damage with membrane lipid peroxidation and oxidization of GSH into GSSG, which destructed the antioxidant defence system of *C. tropicalis* and caused cell death. Under the collective participation of chemical and dynamic antifungal treatment in the killing of *C. tropicalis*, the fact that disruption of biological function for DNA and CPR, metabolic inactivation was displayed. By extending on this base, a battery of chemical biological studies implied that hydroxyethyl naphthalimides should be hopeful to be further exploited as specific antifungal drugs.

## Data Availability

All data are available based on “MDPI Research Data Policies” at https://www.mdpi.com/ethics (accessed on 29 November 2022).

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
