# Peer review of "Identification of Novel Antifungal Skeleton of Hydroxyethyl Naphthalimides with Synergistic Potential for Chemical and Dynamic Treatments"

_molecules, 2022, doi:10.3390/molecules27238453_

Round 1
Reviewer 1 Report
This manuscript describes a significant antifungal potency of novel hydroxyethyl naphthalimides with synergistic potential of chemical and dynamic treatment to combat the fungal resistance.
The result is clear and somewhat interesting. I recommend the acceptance of the paper after changes in some minor issues below:
Line 20 and more:
The word “sulfhydryl” gives the impression that it contains SH groups.
How about writing it in another expression?
Line 91:
Compounds 3a, 6a, and 8a are known compounds. References should be included in the manuscript (and/or supporting information).
Line 164:
You mention its low cytotoxicity. What mechanism do you think gave rise to its selectivity? Please show any comments or citations if you have.
Line 246:
The method for the docking study should be described in Materials and Methods.
Line 257:
You mention pi-sulfur in the Figure 7, do you have any comments or references to cite?
Author Response
(molecules-2033781)
Response to reviewer 1
We would like to express our appreciation to reviewer 1 for suggesting how to improve our manuscript (molecules-2033781). According to the reviewer 1' valuable comments and suggestions as well as the journal-specific guidelines, we have carefully checked our manuscript and revised it point by point. The main changes have been marked with blue fonts highlighted in the revised manuscript. You will find that all these revisions were done according to the suggestions of reviewer 1.
Reviewer 1' Comments to Author:
This manuscript describes a significant antifungal potency of novel hydroxyethyl naphthalimides with synergistic potential of chemical and dynamic treatment to combat the fungal resistance. The result is clear and somewhat interesting. I recommend the acceptance of the paper after changes in some minor issues below:
Response: Thank reviewer very much for valuable comments and recommendation for publication. You will find that all these revisions were done as follows.
Line 20 and more: The word “sulfhydryl” gives the impression that it contains SH groups. How about writing it in another expression?
Response: Thank reviewer very much. According to the reviewer’s suggestion, we have changed “sulfhydryl” into “thioether” in the revised manuscript.
Line 91: Compounds 3a, 6a, and 8a are known compounds. References should be included in the manuscript (and/or supporting information).
Response: Thank reviewer very much. According to the reviewer’s suggestion, we have added the corresponding references in Line 92 in the revised manuscript.
Line 164: You mention its low cytotoxicity. What mechanism do you think gave rise to its selectivity? Please show any comments or citations if you have.
Response: Thank reviewer very much. These compounds could selectively target fungal cell membranes due to an electrostatic distinction on the membranes between fungi and mammalian cells. The corresponding description and references have been added in the revised manuscript.
The added description and references in Lines 171 and 841 in the revised manuscript: These compounds could selectively target fungal cell membranes due to an electrostatic distinction on the membranes between fungi and mammalian cells [33,34].
[33] Lin, S.M.; Wade, J.D.; Liu, S.P. De Novo design of flavonoid-based mimetics of cationic antimicrobial peptides: Discovery, development, and applications. Acc. Chem. Res. 2021, 54, 104−119.
[34] Lin, S.M.; Li, H.X.; Tao, Y.W.; Liu, J.Y.; Yuan, W.C.; Chen, Y.Z.; Liu, Y.; Liu, S.P. In vitro and in vivo evaluation of membrane-active flavone amphiphiles: Semisynthetic kaempferol-derived antimicrobials against drug-resistant gram-positive bacteria. J. Med. Chem. 2020, 63, 5797−5815.
Line 246: The method for the docking study should be described in Materials and Methods.
Response: Thank reviewer very much. According to the reviewer’s suggestion, we have added the method for the docking study in Line 748 in the revised manuscript.
Line 257: You mention pi-sulfur in the Figure 7, do you have any comments or references to cite?
Response: Thank reviewer very much. The interaction of compound 7f with cytochrome P450 reductase was done according to the computer-aided simulation. Moreover, pi-sulfur is also one of the non-covalent interactions. (See reference [a])
[a] Rahman, M.M.; Biswas, S.; Islam, K.J.; Paul, A.S.; Mahato, S.K.; Ali, M.A.; Halim, M.A. Antiviral phytochemicals as potent inhibitors against NS3 protease of dengue virus. Comput. Biol. Med. 2021, 134, 104492.
Reviewer 2 Report
The topic of the manuscript is of high scientific importance. The research is scientifically grounded and performed systematically. In general, the manuscript is well written. However, several issues need to be resolved to improve the quality of the work presented. For example, some minor typographical and grammatical errors throughout the manuscript should be checked and corrected. Furthermore, the resolution of Figure 8 should be improved.
Author Response
(molecules-2033781)
Response to reviewer 2
We would like to express our appreciation to reviewer 2 for suggesting how to improve our manuscript (molecules-2033781). According to the reviewer 2' valuable comments and suggestions as well as the journal-specific guidelines, we have carefully checked our manuscript and revised it point by point. The main changes have been marked with blue fonts highlighted in the revised manuscript. You will find that all these revisions were done according to the suggestions of reviewer 2.
Reviewer 2' Comments to Author:
The topic of the manuscript is of high scientific importance. The research is scientifically grounded and performed systematically. In general, the manuscript is well written. However, several issues need to be resolved to improve the quality of the work presented. For example, some minor typographical and grammatical errors throughout the manuscript should be checked and corrected. Furthermore, the resolution of Figure 8 should be improved.
Response: Thank reviewer very much for valuable comments. We have carefully checked and corrected some errors throughout the manuscript. Moreover, the resolution of Figure 8 has been improved in the revised manuscript.
Reviewer 3 Report
The manuscript "Identification of novel antifungal skeleton of hydroxyethyl naphthalimides with synergistic potential for chemical and dynamic treatments" shows highly interesting results on the strong anti-fungal activity of new compounds for consideration as new drug candidates.
The increase in anti-fungal resistance is of concern, as there are few drugs available which are safe and effective in treating fungal infections in humans.
I am a microbiologist, so I cannot give meaningful comments on the chemistry aspect of this paper.
While I do believe that these compounds seem to have promise, I believe further experiments are warranted before they can be demeaned to be more effective than current compounds, such as fluconazole. One aspect that was immediately clear was the use of a haemolysis assay to determine cell viability. The authors mention that they look a cell viability of LO2 cells, but the methods aren't described in the methods section. Red blood cells are anucleated and are functionally extremely different from other cell types and are not representative of the cells to be treated at the site of infection, such as epithelial cells.
The difference between the viability of a anucleated vs a nucleated cell is very important in this context, as the authors discuss in great detail about the DNA binding affinity of their compounds. Therefore, the authors should either perform cell viability experiments in epithelial cells from organs which are infected by these pathogenic fungi (lung epithelium for A. fumigatus, and skin/mucosa for the candida species), OR they can discuss more about the choice of method in the paper.
Minor comments:
Reference 1 does not support the statement it is written with. The reference is for plant miRNA expression, but the authors are mentioning human and animal fungal infection
While the English writing appears to be of good quality, there are several sentences which are overly complicated confusing. For example, lines 40-42: "For the purpose of solving this huge challenge, expect for the exploitation of new antifungal drugs, it is also a pragmatic tactic to discover new means to heighten the fungicidal effect." In this example the interjection "except for the...", makes the sentence confusing and doesn't necessarily contribute to the reader's understanding. It would aid the reader if the authors consider to simplify the writing.
Author Response
(molecules-2033781)
Response to reviewer 3
We would like to express our appreciation to reviewer 3 for suggesting how to improve our manuscript (molecules-2033781). According to the reviewer 3' valuable comments and suggestions as well as the journal-specific guidelines, we have carefully checked our manuscript and revised it point by point. The main changes have been marked with blue fonts highlighted in the revised manuscript. You will find that all these revisions were done according to the suggestions of reviewer 3.
Reviewer 3' Comments to Author:
The manuscript "Identification of novel antifungal skeleton of hydroxyethyl naphthalimides with synergistic potential for chemical and dynamic treatments" shows highly interesting results on the strong anti-fungal activity of new compounds for consideration as new drug candidates.
Response: Thank reviewer very much for valuable comments.
The increase in anti-fungal resistance is of concern, as there are few drugs available which are safe and effective in treating fungal infections in humans. I am a microbiologist, so I cannot give meaningful comments on the chemistry aspect of this paper.
While I do believe that these compounds seem to have promise, I believe further experiments are warranted before they can be demeaned to be more effective than current compounds, such as fluconazole. One aspect that was immediately clear was the use of a haemolysis assay to determine cell viability. The authors mention that they look a cell viability of LO2 cells, but the methods aren't described in the methods section. Red blood cells are anucleated and are functionally extremely different from other cell types and are not representative of the cells to be treated at the site of infection, such as epithelial cells. The difference between the viability of a anucleated vs a nucleated cell is very important in this context, as the authors discuss in great detail about the DNA binding affinity of their compounds. Therefore, the authors should either perform cell viability experiments in epithelial cells from organs which are infected by these pathogenic fungi (lung epithelium for A. fumigatus, and skin/mucosa for the candida species), OR they can discuss more about the choice of method in the paper.
Response: Thank reviewer very much for valuable comments. Hemolysis assay has been shown in Figure 4B in the previous manuscript. Moreover, the method for cytotoxicity has been added in the revised manuscript.
In fact, our work was aimed at developing novel antifungal compounds, screening their antifungal activity, and preliminarily elucidating their antifungal mechanism. The above behavior of compounds was not studied in detail. But we will specifically investigate this behavior of compounds, which will be reported in subsequent studies.
Minor comments:
Reference 1 does not support the statement it is written with. The reference is for plant miRNA expression, but the authors are mentioning human and animal fungal infection
Response: Thank reviewer very much for valuable comments. According to the reviewer’s suggestion, we have revised the Reference 1 in the revised manuscript.
While the English writing appears to be of good quality, there are several sentences which are overly complicated confusing. For example, lines 40-42: "For the purpose of solving this huge challenge, expect for the exploitation of new antifungal drugs, it is also a pragmatic tactic to discover new means to heighten the fungicidal effect." In this example the interjection "except for the...", makes the sentence confusing and doesn't necessarily contribute to the reader's understanding. It would aid the reader if the authors consider to simplify the writing.
Response: Thank reviewer very much for valuable comments. According to the reviewer’s suggestion, we have revised the sentence in the revised manuscript.
The previous manuscript: For the purpose of solving this huge challenge, expect for the exploitation of new antifungal drugs, it is also a pragmatic tactic to discover new means to heighten the fungicidal effect.
The revised manuscript: For the purpose of solving this huge challenge, it is a pragmatic tactic to discover new means to heighten the fungicidal effects.